# Examining Factors Associated with E-Cigarette Use among Current Smokers

**DOI:** 10.3390/healthcare11182526

**Published:** 2023-09-13

**Authors:** MinHee Park, HyeYoung Song

**Affiliations:** 1Department of Nursing, Wonkwang University, 460 Iksanda-ro, Iksan-si 54538, Jeollabuk-do, Republic of Korea; minipark@wku.ac.kr; 2Department of Nursing, Woosuk University, Samrye-eup, Wanju-gun 55338, Jeonbuk, Republic of Korea

**Keywords:** adults, e-cigarettes, health, smoking

## Abstract

We present a secondary data analysis of the raw data from the eighth Korea National Health and Nutrition Examination Survey (KNHANES). A total of 827 current smokers who responded that they had smoked >5 packs (100 cigarettes) of cigarettes in their lifetime and were currently smoking traditional cigarettes were selected. This study was conducted to identify sociodemographic, smoking-related, and health-related characteristics that influence the use of e-cigarettes in adult smokers. To examine these factors, general characteristics such as age, marital status, education level, and occupation were included in Model 1, while health-related characteristics such as the level of smoking and depression were included in Model 2. In Model 1, age, a high level of education, and working in an office were found to be significantly correlated with e-cigarette use among smokers, while age and working in the office were found to be significantly correlated with e-cigarette use in Model 2. Therefore, e-cigarette use was high among adult smokers of young ages who were office workers. Although evidence is lacking regarding its safety and use as smoking cessation aids, many smokers have been reported to use e-cigarettes as smoking cessation aids, making it necessary to provide accurate information on e-cigarettes.

## 1. Introduction

### 1.1. Rationale for the Study

Since their introduction in Korea in 2011, the use of electronic cigarettes (e-cigarettes) has been popularized. According to the data from the Korea Disease Control and Prevention Agency (KDCA) [1], e-cigarette use in adult women showed a small increase from 0.3% to 1.9% from its first introduction in 2011 to 2022, but the numbers increased from 3.7% to 8.1% in adult men. While the mean smoking rate of e-cigarettes in Koreans was reported to be 11%, the highest rate was found in both male and female adolescents in their 20 s [1]. Overall, the use rate increased from 4.4% in 2013 to 8.4% in men and from 2.9% to 11.7% in women, showing a threefold increase. Globally, Korea ranks second in increased e-cigarette use [2]. Adolescents are the major e-cigarette users in Korea, ranking second among cigarette users globally [1].

The reasons for the continuous increase in interest in e-cigarettes are their similarity to traditional cigarettes and their effect on smoking cessation, which have been actively advertised strategically by e-cigarette manufacturers [3,4]. In fact, even though e-cigarettes have the same legal regulations as traditional cigarettes, the manufacturers advertise that e-cigarettes are different from traditional cigarettes, influencing the perceptions of the consumers [5]. They are similar in that aerosol is inhaled as in traditional cigarettes. Moreover, devices that use liquids that do not contain nicotine have been approved and sold as “smoking cessation aids” [6]. This has misled consumers and led to the belief that e-cigarettes are smoking cessation aids. Following this trend, e-cigarettes came to be used as a replacement for traditional cigarettes or smoking cessation aids, becoming popular among smokers who have difficulty with or refuse smoking cessation. In addition, a meta-analysis on e-cigarettes [7] showed that adults and adolescents exposed to e-cigarettes may start smoking in the future.

E-cigarettes have been advertised as smoking cessation aids from the beginning, even before safety and efficacy were investigated. However, the controversy on safety remains. Respiratory diseases and decreased lung function have been observed in e-cigarette users [8], and there have been reports that potential carcinogens for the respiratory and gastrointestinal systems as well as the skin have been found on the surface of the e-cigarette cartridge [9]. Although the quantities are much lower than the quantity found in traditional cigarettes, toxic substances such as formaldehyde, acetaldehyde, acrolein, toluene, nitrosonornicotine, and 4-(nitrosomethyl-amino)-1-(3-pyridyl)-butanone are still found in e-cigarettes [10]. Indirect smoking may seem like a nonissue since no smoke is produced when using e-cigarettes; however, existing studies have reported on substances found during smoking in the air around e-cigarette users [11]. Considering the dangers of e-cigarettes, in January 2015 [12], the Ministry of Health and Welfare announced that e-cigarettes contain the same carcinogens as cigarettes (regular cigarettes), and they will strictly crack down on advertising that e-cigarettes have an aiding effect on smoking cessation.

E-cigarette-related factors of adult smokers include demographic characteristics [12], health status [13,14], mental health [14], and smoking characteristics [5]. Health characteristics are important in health research as independent predictors of disease or total mortality [15]. Health characteristics can be identified as lifestyle habits, such as physical activity, obesity, smoking, unhealthy diet, drinking, and genetic factors [15]. As a result, identifying the health status and smoking characteristics of adult e-cigarette smokers can predict the health level of e-cigarette smokers in a state where the harmfulness and safety of e-cigarettes are not clearly confirmed [13]. In addition, by identifying the factors affecting health status related to e-cigarette use in adult smokers, it can be used as basic data to establish health promotion strategies to induce smoking cessation. 

Studies have been conducted regarding e-cigarette use in Korean adults [16,17,18,19,20,21,22]; however, only a few existing studies have analyzed e-cigarette use-related factors in smokers based on smoking-related and health-related characteristics. Therefore, to investigate the characteristics of smokers who use e-cigarettes, the present study aimed to examine sociodemographic, smoking-related, and health-related characteristics as influencing factors for e-cigarette use in smokers. This study may be useful in establishing a smoking cessation intervention strategy by identifying the characteristics of the high-risk group, comprising those who use both traditional cigarettes and e-cigarettes.

### 1.2. Study Objectives

To identify factors influencing e-cigarette use in adult smokers and to identify differences in e-cigarette use among smokers according to the sociodemographic, smoking-related, and health-related characteristics.

## 2. Materials and Methods

### 2.1. Study Design

This descriptive study was conducted to identify factors influencing e-cigarette use in smokers. It is a secondary data analysis study using the data from KNHANES. The KNHANES is a nationwide survey that calculates statistics on a national level in Korea and is used to develop new health policies through the evaluation of the public’s health level.

### 2.2. Study Participants

The 2021 KNHANES raw data [23] provided by the KDCA was used in this study. Of the 7090 respondents of the 2021 KNHANES, 827 current smokers were selected as subjects for this study. Current smokers are those who have smoked more than 5 packs (100 cigarettes) of cigarettes in their lifetime and answered that they currently smoke cigarettes.

### 2.3. Measures

Among the KNHANES variables, general characteristics and some of the smoking and health-related questions were used in this study.

#### 2.3.1. General Characteristics 

General characteristics included sex (male, female), age (20 s, 30 s, 40 s, 50 s, and above 60 s), area of residence (city and rural area), marital status (single, living with spouse, and widowed, widower, divorced), education level (“did not graduate elementary school” and “middle school and above”), household income (low, mid-low, mid-high, and high), and occupation (managers and experts; service and sales workers; agriculture, forestry, and fishery workers; technicians, machine operators, and assembly workers; simple laborers; and unemployed (housewife, student, etc.)).

#### 2.3.2. Smoking-Related and Health-Related Characteristics 

To identify smoking-related characteristics in the participants, the amount of smokingplan for smoking cessation, and experience with attempts to quit smoking were surveyed. For health-related characteristics, the frequency of drinking, stress perception, experience with depression, and subjective well-being were surveyed.

The level of smoking was categorized as “<10 per day”, “>10 per day”, “<20 per day”, and “>20 per day”. Smoking cessation plans were categorized as “no plans to quit smoking”, “someday but not within 6 months”, “within 6 months”, or “within 1 month” as answers to the question “Do you plan to quit smoking within the next 1 month?” Experience with attempts to quit smoking were categorized as “Yes” and “no” as answers to the question “Have you quit smoking for at least a day (24 h) in the past 1 year?” 

Drinking frequency was categorized as “not at all”, “less than once a month”, “2–3 times a month”, or “more than 2 times a month” as answers to the question “How often did you drink in the past 1 year?” Stress perception was categorized as “feel less” and “feel a lot” as answers to the question “How much stress do you feel in typical daily life?” Experience with depression was categorized “Yes” and “No” as answers to the question “Have you felt depressed or despair to the point that it interfered with daily life for more than 2 consecutive weeks in the past 1 year?” Subjective well-being was categorized as “good”, “moderate”, and “bad” as answers to the question “what do you think of your subjective health status?”

#### 2.3.3. Use of E-Cigarettes

Current use of e-cigarettes, which was the dependent variable of this study, was defined as responding “yes” to the question “have you used vapes in the past month?” or responding “use every day” and “use sometimes” to the question “are you currently using e-cigarettes (examples of heated cigarettes, IQOS, GLO, Lil, etc.)?” 

### 2.4. Data Collection

To ensure the use of a representative sample, some of the KNHANES raw data processed with the stratified colony probability extraction method and the phylogenetic extraction method were analyzed secondarily in this study. Among KNHANES items, data from health and nutritional surveys were used, which were collected through face-to-face interviews or the self-completion method during house visits. KNHANES involved an expert survey team including nurses, nutritionists, and public health majors who were qualified based on regular training and on-site quality control [23]. The data used in the study were raw anonymized data from the KNHANES website (https://knhanes.cdc.go.kr (accessed on 11 August 2023)). 

### 2.5. Data Analysis

For data analysis, a complex sample analysis was conducted in accordance with the “KNHANES raw data analysis guidelines” using SPSS/Win 24.0. The KNHANES sampling was conducted according to the complex sample design method. Therefore, in order to increase the accuracy of the estimation, a complex sample analysis method was used to reflect stratification, cluster, and weight. According to the general and smoking and health-related characteristics, the following analyses were conducted: weighted percentages, means, and standard deviations; the complex sample t-test and complex sample χ^2^-test to evaluate differences in e-cigarette use; and complex sample multiple regression analysis to identify factors influencing the use of e-cigarettes. Statistical significance was set at *p* < 0.05.

## 3. Results

### 3.1. Differences in E-Cigarette Use According to Sociodemographic Characteristics of Smokers

The Chi-square test was conducted to verify whether there were differences in e-cigarette use according to the general characteristics of the smokers (Table 1). Significant differences were found according to age (χ^2^ = 36.68, *p* < 0.001), marital status (χ^2^ = 20.33, *p* < 0.001), education level (χ^2^ = 21.97, *p* < 0.001), and occupation (χ^2^ = 39.31, *p* < 0.001).

Upon examining e-cigarette usage according to the general characteristics that showed significant differences, the use of e-cigarettes was high in the younger age groups, as the use rate was found to be 26.1% in the 20 s, 23.7% in the 30 s, 16.9% in the 40 s, and 7.3% in the 50 s age groups. The use of e-cigarettes was higher in unmarried smokers (24.6% single vs. 12.3% married). Regarding education level, higher education level was associated with high usage of e-cigarettes, with a rate of 3.4% for middle school graduates or below, 17.0% for high school graduates, and 21.5% for college graduates and above. Regarding occupation, the use rate was the highest among officer workers, followed by service and sales workers, managers and experts, technicians, unemployed, and simple laborers (33.3%, 22.6%, 19.8%, 12.8%, 10.8%, and 6.8%, respectively).

### 3.2. Differences in Current E-Cigarette Use According to Smoking-Related and Health-Related Characteristics of Smokers

The chi-square test results showed significant differences in e-cigarette usage according to level of smoking (χ^2^ = 10.83, *p* = 0.031) and depression (χ^2^ = 7.21, *p* = 0.011) (Table 2). 

Furthermore, those who smoked less were more likely to use e-cigarettes, as the usage rate was 22.7% in those who smoked <10 per day, 14.4% in those who smoked 10–20 per day, and 7.9% in those who smoked >20. In addition, e-cigarette usage was higher in participants without depression, as the usage rate was 7.3% in those with depression and 17.7% in those without depression.

### 3.3. Factors Related to Current E-Cigarette Use in Smokers

Multivariate logistic regression analysis was conducted to verify the factors influencing e-cigarette use in smokers (Table 3), inputting significant variables from the chi-square results in Table 1 and Table 2 as independent variables.

Using non-users of e-cigarettes as the reference group, the factors that affected the possibility of e-cigarette use were verified, and the significant variables found in the previous chi-square test were substituted as independent variables. In Model 1, the general characteristics variables including age, marital status, education level, and occupation were used, while health-related characteristics including smoking level and depression were used in Model 2. 

In Model 1, in which general characteristics were included as variables, age was a significant factor, as higher age was significantly associated with a lower possibility of e-cigarette use (OR = 0.96, *p* < 0.001). Regarding education level, the possibility of e-cigarette use was significantly higher in college graduates or higher, compared with below middle school graduates (OR = 3.31, *p* = 0.043). Regarding occupation, e-cigarette usage was significantly higher among office workers compared with managers and experts (OR = 2.51, *p* = 0.024). 

When Model 2 was verified by including smoking level and depression, which were health-related variables, no significant association was found among the health-related characteristics. However, for the general characteristics, age and occupation were significant factors as in Model 1, as higher age was associated with a significantly lower possibility of e-cigarette use (OR = 0.96, *p* < 0.001), and office workers were significantly more likely to use e-cigarettes compared with managers and experts (OR = 2.46, *p* = 0.030).

## 4. Discussion

This was a secondary analysis study that used the raw data from the eighth KNHANES [23] that was conducted to identify sociodemographic, smoking-related, and health-related characteristics influencing e-cigarette usage in adult smokers. To identify the influencing factors for e-cigarette use in smokers, general characteristics such as age, marital status, education level, and occupation were included in Model 1, and health-related characteristics such as smoking level and depression were included in Model 2. Model 1 showed that age, a high level of education, and being an office worker were significantly associated with e-cigarette usage in smokers, while Model 2 showed that age and being an office worker were significantly associated with e-cigarette usage.

E-cigarette usage was found to be higher in the younger age groups, consistent with the finding from an existing study that e-cigarette usage increased in younger age groups. This is explained by the fact that younger e-cigarette users are more likely to be novelty seekers than the elderly and are more likely to be influenced by the people around them [24] or that they are exposed to online marketing through social media [25].

A higher education level was associated with higher usage of e-cigarettes. Results from previous studies have also shown that individuals with higher incomes who were less likely to avoid risks, were working in offices, had fewer members in their households, were of younger ages, and had a higher level of education were more likely to use e-cigarettes [26]. This shows that e-cigarettes are accessible to those with relatively high socioeconomic status. 

E-cigarette usage was found to be high in adults who were office workers. An explanation for this may be that those working in an office try to avoid the traditional cigarette smell due to working in an enclosed space more than those working outdoors [24]. It has been reported that smokers are more likely to use e-cigarettes if they believe that e-cigarettes may have a reduced unpleasant smell, compared with traditional cigarettes [27].

Although the multivariate analysis in this study did not show significant differences, the variable that showed significant differences in e-cigarette usage according to general characteristics was being married. E-cigarette usage was found to be higher for married smokers than for single smokers. This can be explained by the fact that married smokers tend to think of their health as well as the damage from secondary smoking after forming a family. They believe that e-cigarettes are less harmful than traditional cigarettes and are helpful for health and smoking cessation [6], using them as cigarette replacements to reduce harm to themselves and their family members.

When e-cigarette usage according to smoking-related characteristics in adult smokers was examined, significant differences were observed according to smoking level. When fewer cigarettes were smoked, e-cigarette usage was high. This is similar to the finding that e-cigarettes are used as a replacement for traditional cigarettes [7] and that groups that smoke less are more likely to use e-cigarettes.

When examining e-cigarette usage according to health-related characteristics in adult smokers, significant differences were observed in depression, as participants without depression had higher e-cigarette use rates than those with depression. A previous study reported that higher depression scores were associated with a higher likelihood of using e-cigarettes and a higher likelihood of dual use with traditional cigarettes in past smokers compared with current smokers [20]. Nicotine is known to affect many other neurotransmitters in the central nervous system (CNS), such as serotonin [28]. Chronic nicotine exposure may affect depressive symptoms by reducing serotonin levels [29]. According to the serotonin theory of depression, impaired serotonin function can affect mood and lead to clinical depression [30]. Assessing current smokers for depressive symptoms helps prevent withdrawal symptoms. Depressive symptoms are an obstacle to the success of smoking cessation interventions [31]. Longitudinal studies are needed to clearly confirm the causal relationship between e-cigarette use and depressive symptoms.

In the present study, e-cigarette use rates were higher in the younger age groups (20 s and 30 s) and in office workers; therefore, these age groups are more susceptible to the risks associated with using e-cigarettes. Therefore, appropriate counseling and education on the risks of dual smoking as well as the hazards of e-cigarettes must be provided. Despite the controversies on the safety and efficacy of smoking cessation for e-cigarettes, many smokers tend to believe that e-cigarettes help with smoking cessation or are less harmful. Therefore, the perception of the public regarding e-cigarettes should be improved. To deliver accurate information, the hazards of e-cigarettes as well as the risk of dual smoking should be emphasized when promoting smoking cessation.

This study analyzed large-scale sample group data using the national statistics from KNHANES and is advantageous in that it includes a representative sample. However, this is a cross-sectional study using survey results, and causal relationships could not be determined. In addition, as a survey study, it may have been affected by reporting bias. In this study, e-cigarette use questionnaires did not include information related to the e-cigarette use period or the nicotine concentration in e-cigarettes. In addition, since the research subject was limited to one year, the research sample on e-cigarette users could not be sufficiently collected. It seems that a follow-up study based on the accumulated follow-up data with the addition of these survey items will be required in the future.

## 5. Conclusions

This study was conducted to investigate the sociodemographic, smoking-related, and health-related characteristics of smokers who use e-cigarettes using the survey data from the KNHANES. E-cigarette use was high among adult smokers with young age who were office workers. Although evidence is lacking regarding the safety of e-cigarettes and its efficacy as a smoking cessation aid, many smokers still believe e-cigarettes are smoking cessation aids. It seems that we need to prepare a customized e-cigarette smoking cessation policy for young office workers who use e-cigarettes.

## Figures and Tables

**Table 1 healthcare-11-02526-t001:** Differences in current E-cigarette use according to general characteristics of smokers.

Variables	Total	Current E-Cigarette Use		χ^2^	*p*-Value
		Yes	No		
Total	827 (100.0)	112 (16.4)	715 (83.6)		
Sex					
Male	678 (83.5)	91 (16.3)	587 (83.7)	0.04	0.864
Female	149 (16.5)	21 (17.0)	128 (83.0)		
Age					
19–29	129 (21.3)	35 (26.1)	94 (73.9)	36.68	<0.001
30–39	111 (17.6)	24 (23.7)	87 (76.3)		
40–49	194 (23.2)	35 (16.9)	159 (83.1)		
≥50	393 (37.9)	18 (7.3)	375 (92.7)		
Residence					
City	651 (85.1)	94 (17.1)	557 (82.9)	1.35	0.280
Rural area	176 (14.9)	18 (12.8)	158 (87.2)		
Marital status					
Married	602 (66.2)	61 (12.3)	541 (87.7)	20.33	<0.001
Unmarried	225 (33.8)	51 (24.6)	174 (75.4)		
Education level					
≤Middle school	196 (16.0)	4 (3.4)	192 (96.6)	21.97	<0.001
High school	360 (47.8)	54 (17.0)	306 (83.0)		
≥College	271 (36.2)	54 (21.5)	217 (78.5)		
Household income					
Low	130 (11.2)	4 (4.4)	126 (95.6)	12.30	0.067
Mid-low	232 (26.2)	31 (16.3)	201 (83.7)		
Mid-high	242 (33.1)	39 (17.4)	203 (82.6)		
High	223 (29.5)	38 (20.0)	185 (80.0)		
Occupation					
Manager and experts	100 (12.8)	22 (19.8)	78 (80.2)	39.31	<0.001
Office workers	84 (11.8)	24 (33.3)	60 (66.7)		
Service and sales workers	126 (17.1)	24 (22.6)	102 (77.4)		
Technician	139 (17.8)	16 (12.8)	123 (87.2)		
Simple laborers	137 (13.5)	6 (6.8)	131 (93.2)		
Unemployed	241 (27.0)	20 (10.8)	221 (89.2)		

Analyzed by Rao–Scott chi-squared analysis; values are presented as numbers (weighted%).

**Table 2 healthcare-11-02526-t002:** Differences in current e-cigarette use according to health characteristics of smokers.

Variables	Total	Current E-Cigarette Use		χ^2^	*p*-Value
		Yes	No		
Total	827 (100.0)	112 (16.4)	715 (83.6)		
Amount of smoking					
<10 per day	229 (28.4)	43 (22.7)	186 (77.3)	10.83	0.031
10–20 per day	546 (66.1)	67 (14.4)	479 (85.6)		
>20 per day	52 (5.5)	2 (7.9)	50 (92.1)		
Smoking cessation plans					
Never	321 (39.1)	50 (20.6)	271 (79.4)	6.78	0.168
More than 6 months	291 (36.2)	35 (13.7)	256 (86.3)		
Within 6 months	79 (9.0)	12 (14.8)	67 (85.2)		
Within 1 month	136 (15.7)	15 (13.3)	121 (86.7)		
Quit smoking trial					
Yes	386 (46.1)	46 (13.5)	340 (86.5)	4.27	0.081
No	441 (53.9)	66 (18.9)	375 (81.1)		
Frequency of drinking					
None	122 (11.8)	9 (11.3)	113 (88.7)	2.40	0.626
≤1 per month	176 (22.3)	27 (18.4)	149 (81.6)		
2–4 per month	205 (25.8)	27 (16.9)	178 (83.1)		
≥2 per week	324 (40.1)	49 (16.5)	275 (83.5)		
Perceived stress					
Feel a lot	280 (33.5)	48 (20.2)	232 (79.8)	4.28	0.090
Feel less	547 (66.5)	64 (14.5)	483 (85.5)		
Depressed experience					
Yes	118 (12.5)	8 (7.3)	110 (92.7)	7.21	0.011
No	709 (87.5)	104 (17.7)	605 (82.3)		
Subjective health status					
Good	241 (30.2)	38 (19.5)	203 (80.5)	4.65	0.213
Moderate	404 (51.1)	59 (16.5)	345 (83.5)		
Bad	182 (18.7)	15 (11.3)	167 (88.7)		

Analyzed by Rao–Scott chi-squared analysis; values are presented as numbers (weighted%).

**Table 3 healthcare-11-02526-t003:** Factors affecting current e-cigarette use among smokers.

Variables	Model 1			Model 2		
	OR	95% CI	*p*-Value	OR	95% CI	*p*-Value
Age	0.96	0.94–0.98	<0.001	0.96	0.94–0.98	<0.001
Marital status						
Unmarried	1.00			1.00		
Married	0.86	0.44–1.66	0.646	0.86	0.44–1.69	0.667
Education						
≤Middle school	1.00			1.00		
High school	2.79	0.87–9.02	0.085	2.68	0.85–8.51	0.093
≥College	3.31	1.04–10.54	0.043	3.05	0.97–9.66	0.057
Occupation						
Manager and experts	1.00			1.00		
Office workers	2.51	1.13–5.58	0.024	2.46	1.09–5.52	0.030
Service and sales workers	1.17	0.49–2.82	0.718	1.17	0.48–2.82	0.733
Technicians	0.88	0.38–2.03	0.758	0.92	0.39–2.18	0.853
Simple laborers	0.55	0.18–1.66	0.288	0.56	0.18–1.70	0.305
Unemployed	0.63	0.26–1.56	0.316	0.67	0.26–1.68	0.387
Amount of smoking						
<10				1.00		
10–20				0.82	0.52–1.29	0.394
>20				0.64	0.14–2.83	0.553
Depressed experience						
No				1.00		
Yes				0.45	0.20–1.00	0.051

Analyzed by multivariate logistic regression analysis; OR, Odds Ratio; CI, Confidence Interval.

## Data Availability

Not applicable.

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
