# Peer review of "Examining Factors Associated with E-Cigarette Use among Current Smokers"

_healthcare, 2023, doi:10.3390/healthcare11182526_

Round 1
Reviewer 1 Report
This is a prospective study that analyzed the Relationship among adult’s smoking, health-related characteristics and e-cigarette use from the eighth Korea National Health and Nutrition Examination Survey (KNHANES). In the study (survey), 7,090 respondents, 881 current smokers who responded that they had smoked cigarettes and 89 major variables were selected as the study participants. To examine these factors, general characteristics such as age, marital status, education level, and occupation were included in Model 1, while health-related characteristics such as level of smoking and depression were included in Model 2. The authors claim that e-cigarettes have efficacy as smoking cessation aid and smokers believe that e-cigarettes are smoking cessation aid.
Major comments:
1. Although this is a good data analysis the author only considers 9% of the participants, which reflects trial data (small sample size for data inferring).
2. The data provided by the author indicates cigarette consumption and does not include e-cigarettes (2.2. Study Participants; lines 86-90).
3. The author didn’t use a rational measure which is indicated as major variables.
4. Although the author stratified the data, it does show the relationship between efficacy as smoke cessation aid.
5. What kind of analysis did the author make?
6. What are the characteristics for e-cigarettes user and regular cigarettes users?
7. Variables for both types of consumers? Do they both? Demographics against users? Friendly smoker, group smoker? Use only cigarettes or both?
8. Behavior for aid e-cigarette smoker?
9. What is statistical analysis?
Introduction
Too long and not a real description of the work, the authors the authors use a conclusion for an introduction. No real rational of the work or why they include all this review.
Material and methods
The materials and methods section needs further refinement and organization, as its current complexity impedes comprehension and readability.
Discussion and conclusion
The conclusion section has extended considerably in length, yet it regrettably lacks substantive content, resulting in a tedious narrative devoid of meaningful outcomes or conclusive insights.
The manuscript contains some running and a minor revision is needed for improving clarity and coherence.
Author Response
We thank the reviewers for their helpful comments. We believe that the revised version of our manuscript is much improved as a result of their insightful suggestions.
Below, we provide point-by-point responses to the reviewers’ comments, describing our revisions in detail. Thank you for your consideration.
* In the manuscript, the revised sections are highlighted in red.
We thank the reviewers for their helpful comments. We believe that the revised version of our manuscript is much improved as a result of their insightful suggestions.
Below, we provide point-by-point responses to the reviewers’ comments, describing our revisions in detail. Thank you for your consideration.
* In the manuscript, the revised sections are highlighted in red.

Reviewer 2 Report
The manuscript assessed factors associated e-cigarette use among smokers using data from the eighth KNHANES. I have a few comments and questions.
Title & Abstract
Suggest revising the title to reflect the key aim of the manuscript that is to examine the characteristics of e-cigarette users among current adult smokers. Suggest including the analytic sample in the abstract.
Introduction
Line 41. E-cigarettes produce an “aerosol” not “gas”.
Lines 41-46. The authors alleged that dual cigarette and e-cigarette users are at high risk. But this has not been discussed in the introduction at all. To my knowledge, this is yet conclusive evidence regarding whether dual use confers more, less, or similar health risk compared with exclusive cigarette smoking. There is a general consensus that exclusive e-cigarette use is less harmful compared with exclusive cigarette smoking, this should be made clear in the introduction or in the discussion section.
Materials and methods
Although KNHANES is a nationally representative sample, the analytic sample size was only 827. The small sample size clearly has an implication to the precision of the estimations, as many associations appeared to be marginally significant. One of my key questions is why the authors did not pool multiple waves of KNHANES data for the present study? Since e-cigarette use questions are available in many previous waves as well.
Lines 109-111. The authors described the measure of nicotine dependence, but it was not included in Table 2. This is quite strange.
Data analysis. The description of statistical analyses performed is somewhat unconventional. “Complex sample multiple regression analysis…” should probably be something like “Multivariable logistic regression was used to assess factors associated with e-cigarette use.” Please specify how complex survey design parameters, such as sample weight, PSU, strata, were incorporated in the analysis.
When examining factors associated with e-cigarette use, the authors retained those appeared statistically significant in univariate analyses. Please clarify this technical detail and defend the choice of including predictors based only on statistical test. Although sex was not significantly associated with e-cigarette use, I strongly believe that it should be included in the multivariable regression model, because it is a key sociodemographic characteristic.
Results
Lines 175-176. “The observed values for the dependent variables for all independent variables confirmed normal distribution through normal p-p curve”. I am having a hard time understand this statement – the dependent variable was a binary variable how could it have a normal distribution?
Discussion
Lines 251-253. The reasons for the contradictory finding on depression may or may not be the problem of reverse causation. Many previous studies also used cross-sectional data, even using the same data source KNHANES found a higher risk of depression associated with e-cigarette use (e.g., You et al, 2023; Lee et al 2020; Lee & Lee 2019). A better explanation is needed given previous evidence.
Lines 267 and 271. Why is lacking duration of e-cigarette use or concentration of nicotine a limitation for the present study of assessing factors associated with e-cigarette use?
References
You, M.A., Choi, J. and Son, Y.J., 2023. Associations of dual use of tobacco cigarettes and e‐cigarettes, sleep duration, physical activity and depressive symptoms among middle‐aged and older Korean adults. Nursing Open, 10(6), pp.4071-4082.
Lee, S., Oh, Y., Kim, H., Kong, M. and Moon, J., 2020. Implications of electronic cigarette use for depressive mood: a nationwide cross-sectional study. Medicine, 99(40).
Careful proofreading is needed.
Author Response
We thank the reviewers for their helpful comments. We believe that the revised version of our manuscript is much improved as a result of their insightful suggestions.
Below, we provide point-by-point responses to the reviewers’ comments, describing our revisions in detail. Thank you for your consideration.
* In the manuscript, the revised sections are highlighted in red.

Reviewer 3 Report
Dear authors,
Kindly find my suggestion for your consideration.
Background: Well-written, however, I would like to suggest including justification for the selection of independent variables in the background.
Objective: Clear and concise, however, suggest changing the title according to the objective. Furthermore, the word “relationship” is not appropriate due to the limitations of the sampling technique.
Methodology: To justify deciding on the participants that had smoked >5 packs (100 cigarettes) of cigarettes in their lifetime. To clarify if there are any confounding factors in this study, refer to this study (Green et al, 2020) https://bmcpublichealth.biomedcentral.com/articles/10.1186/s12889-020-8270-3
Results & Discussion: Analysis and interpretation as well as discussion were done appropriately
Conclusion: Clear, but may need to change the manuscript’s title
Author Response

(The authors gave the same response as above.)

Round 2
Reviewer 1 Report
no comments
no comments
Author Response
We wanted to improve English quality by entrusting the English translation to Editage

Reviewer 2 Report
I appreciate the opportunity to evaluate the revised manuscript. I have a few remaining questions.
My most important question was (and still is) the small analytic sample size (n=827), which clearly has an implication to the precision of the estimations, as many associations appeared to be marginally significant. I still want to know why the authors did not pool multiple waves of KNHANES data for the present study, given e-cigarette use questions are available in many previous waves as well?! It is inadequate just to state this as a limitation. I could see no reason why this can not be done.
The revised title of the study still does not explicitly state the key aim of the study, i.e., examining factors associated with e-cigarette use among current smokers.
No comment.
Author Response
We have attached our response to reviewer
